# Generating Salt-Affected Irrigated Cropland Map in an Arid and Semi-Arid Region Using Multi-Sensor Remote Sensing Data

Deji Wuyun [1,2] , Junwei Bao [1], Luís Guilherme Teixeira Crusiol [3], Tuya Wulan [1], Liang Sun [2] , Shangrong Wu [2], Qingqiang Xin [1], Zheng Sun [2], Ruiqing Chen [2], Jingyu Peng [4], Hongtao Xu [5], Nitu Wu [6], Anhong Hou [1], Lan Wu [7] and Tingting Ren [1,8,*]

1   Research Center of Agricultural Remote Sensing Engineering Technology in Inner Mongolia Autonomous Region, Institute of Rural Economic and Information, Inner Mongolia Academy of Agricultural & Animal Husbandry Sciences, Hohhot 010031, China
2   Institute of Agricultural Resources and Regional Planning, Chinese Academy of Agricultural Sciences, Beijing 100081, China
3   Embrapa Soja (National Soybean Research Center-Brazilian Agricultural Research Corporation), Londrina 86001-970, Brazil
4   Institute of Resources, Environment, Sustainable Development, Inner Mongolia Academy of Agricultural & Animal Husbandry Sciences, Hohhot 010031, China
5   Institute of Grassland Research, Chinese Academy of Agricultural Sciences, Hohhot 010010, China
6   Key Laboratory of Grassland Resources of the Ministry of Education, College of Grassland, Resources and Environment, Inner Mongolia Agricultural University, Hohhot 010011, China
7   College of Resources and Environmental Economics, Inner Mongolia University of Finance and Economics, Hohhot 010070, China
8   Asia Hub, Nanjing Agricultural University, Nanjing 210095, China
*   Correspondence: 2018201089@njau.edu.cn

**Abstract:** Soil salinization is a widespread environmental hazard and a major abiotic constraint affecting global food production and threatening food security. Salt-affected cropland is widely distributed in China, and the problem of salinization in the Hetao Irrigation District (HID) in the Inner Mongolia Autonomous Region is particularly prominent. The salt-affected soil in Inner Mongolia is 1.75 million hectares, accounting for 14.8% of the total land. Therefore, mapping saline cropland in the irrigation district of Inner Mongolia could evaluate the impacts of cropland soil salinization on the environment and food security. This study hypothesized that a reasonably accurate regional map of salt-affected cropland would result from a ground sampling approach based on PlanetScope images and the methodology developed by Sentinel multi-sensor images employing the machine learning algorithm in the cloud computing platform. Thus, a model was developed to create the salt-affected cropland map of HID in 2021 based on the modified cropland base map, valid saline and non-saline samples through consistency testing, and various spectral parameters, such as reflectance bands, published salinity indices, vegetation indices, and texture information. Additionally, multi-sensor data of Sentinel from dry and wet seasons were used to determine the best solution for mapping saline cropland. The results imply that combining the Sentinel-1 and Sentinel-2 data could map the soil salinity in HID during the dry season with reasonable accuracy and close to real time. Then, the indicators derived from the confusion matrix were used to validate the established model. As a result, the combined dataset, which included reflectance bands, spectral indices, vertical transmit–vertical receive (VV) and vertical transmit–horizontal receive (VH) polarization, and texture information, outperformed the highest overall accuracy at 0.8938, while the F1 scores for saline cropland and non-saline cropland are 0.8687 and 0.9109, respectively. According to the analyses conducted for this study, salt-affected cropland can be detected more accurately during the dry season by using just Sentinel images from March to April. The findings of this study provide a clear explanation of the efficiency and standardization of salt-affected cropland mapping in arid and semi-arid regions, with significant potential for applicability outside the current study area.

**Keywords:** irrigation district; cropland; quantile and quantile plots testing; dry season; Google Earth Engine

## 1. Introduction

Soil salinization is a matter of concern in agriculture, as the excess salt hinders crop growth by obstructing the ability to uptake water. In another sense, it causes a loss in soil fertility and leads to the desertification of cropland [1,2]. According to the estimation released by the Food and Agriculture Organization (FAO), there are more than 424 million hectares of topsoil (0–30 cm) and 833 million hectares of subsoil (30–100 cm) are salt-affected around the globe (8.7% of the planet) [3]. Most of them can be found in naturally arid or semi-arid environments in Africa, Asia and Latin America [4]. Soils are easily affected by salt in arid and semi-arid regions where low rainfall and high evapotranspiration lead to the concentration of salts such as sodium, magnesium and calcium to form saline soils [5–8]. FAO launched the Global Map of Salt-Affected Soils in 2021, although the salt-affected soil of China has not been included in that. Nonetheless, estimates show that 20 to 50% of irrigated soils across all continents are too salty, implying that over 1.5 billion people face significant challenges in meeting rising food demand due to severe cropland salinity and cropland degradation [9].

Saline cropland is an essential part of reserve cropland in the Inner Mongolia Autonomous Region in China and is an integral part of the cropland restoration program [10]. The salt-affect soil in the Inner Mongolia Autonomous Region is mainly disturbed in the Xiliao River Plain in the east and Hetao Irrigation District (HID) in the west. The cropland of HID is dominated by saline soil and accounts for 30.5% of the saline cropland in Inner Mongolia [10]. In the early stage of the reclamation HID, flood irrigation without drainage facilities caused the secondary salinization of the field soil. For now, cropland salinization has gradually evolved into the main factor restricting the sustainable development of agriculture in HID. Therefore, the severe salinity cropland is a typical area for the agricultural management department's soil rehabilitation program, which has attracted the interest of many academics [11,12].

The cropland soil salinity in HID is mainly adapted from the irrigation water of the Yellow River. Only 20% of the initial salt can be discharged through drainage, while 80% of the salt is kept in the soil of the irrigation area, showing a salinization trend [13]. Soil salinity will adversely affect plant growth, crop yields, and underground water quality, leading to soil erosion and land degradation [14]. The hazard of soil salinity is not limited to the environment but also includes the economy. For example, for the secondary salinization of the land in the Sultanate of Oman, the direct economic loss from mild to moderate salinity is about 1604 US dollars per hectare, and the direct economic loss from mild to severe salinity is as high as 4352 US dollars per hectare [15]. Thus, knowing the spatial distribution of salt-affected cropland is an urgent need to alleviate the contradiction between humans and land [16,17], which is also vital for promoting the high-quality development of the national agricultural economy [18]. At the same time, the eradicate because of dynamic and accessible restress from salinization after agricultural activities seriously endangers the sustainable development of agriculture and its productivity, which makes the timely detection of salt-affected cropland within HID with limited cropland resources particularly urgent [19–21].

Traditionally, soil salinity was measured by collecting soil samples and analyzing them in a laboratory to determine their solute concentration or electronic conductivity [22]. However, due to intensive sampling being time-consuming and expensive, the spatial variability of soil salinity is hardly fully characterized traditionally in a large area. Remote sensing data and techniques can more effectively provide economic and rapid tools and methods for mapping soil salinity [23]. Remote sensing data and its analyzing processes have gradually become the most convenient method of mapping soil salinity since black-

and-white and color aerial photographs were used to describe salinity-stressed soils in the 1960s. Multispectral imagery such as Landsat [24], Sentinel [25], IKONOS [26], Quick-Bird [27] and UAV-Borne [28] are highly suitable for evaluating soil salinity. In the last three decades of research on monitoring saline soils, multispectral sensors have been mainly used. In addition, some researchers have emphasized the importance of ground sample data [29,30].

In practical applications, multispectral sensors also show limitations, as their spectral resolution and fewer bands affect the quality and quantity of information provided. Many current studies pointed out this limitation, thus monitoring the salinity using hyperspectral [31] and thermal infrared data [32], even Synthetic Aperture Radar (SAR) data [33] in the last few years. Nevertheless, the broad acquisition capability of Sentinel data, high spatial resolution (10 m), and the combination of active and passive remote sensing data can compensate for the deficiencies of multispectral data widely available for free. Remote sensing data with meter-level high resolution or sub-meter-level resolution (IKONOS, QuickBird, WroldView-2, GF series) have also been gradually introduced into salinity mapping research and have become indispensable data sources. Mapping the salinity of cropland combining high spatial resolution images and ground sampling data using machine learning algorithms is mainly carried out at the field scale or farm scale [34]. However, the validity and reliability of such a method need to be assessed in a larger area.

Recent years have seen an increase in nonparametric machine learning techniques, particularly Random Forest (RF), to calculate soil salinity [35,36]. Since it can manage the high dimensionality and multicollinearity of remote sensing data with excellent classification accuracy and insensitivity to overfitting, RF is one of the most extensively used algorithms in land cover classification. Additionally, it has been stated that RF in the Google Earth Engine (GEE) platform provides unassailable benefits in the remote sensing classification of land cover in a large area [37,38]. Some researchers have demonstrated that RF outperforms other popular nonparametric machine learning algorithms, which can significantly increase soil salinity mapping accuracy [25,39].

However, many scholars have shown that using remote sensing technology to map cropland salinity in arid and semi-arid regions is challenging [23,24,40]. It is mainly because the bare ground and other sparse vegetation are easily confused with saline soil in spectral reflectance [33,41]. Alternatively, the method based on spectral reflectance may lead to unreliable results when the soil is moisturizing or the soil salts are not exposed on the soil surface in crystalline form but mixed with other soil components [42]. In this case, SAR data, frequently employed in detecting soil salinity, can capture information that is challenging to acquire using multispectral imagery. Various remote sensing data have already been used to study saline soil in HID. Nonetheless, the majority of these studies have focused on single sensors rather than multi-sensor images. Therefore, to comprehend the main mechanism causing agricultural salinization and degradation, a salt-affected map using a wide range of remote sensing data must be acquired in almost real time.

To fill this gap, the following questions will be addressed in this study: Is the PlanetScope image of April appropriate for sample collection employing the Visual Interpretation strategy? If so, how can the samples' validity—which includes cropland that is both saline and non-saline—be estimated? How to quickly and efficiently map salinized cropland using Sentinel-1 and Sentinel-2 data freely available in GEE? These questions are unavoidable in multi-sensor data-based mapping of salinized cropland, and addressing them is the primary goal of the current study.

The specific objectives of this research are to:

1. Create a cropland base map using global land cover data from ESA WorldCover while masking off roads and irrigation ditches collected from the electronic map of HID;
2. Evaluate the validity of samples, comprising both saline and non-saline cropland, using the quantile and quantile plots testing method;
3. Create a multi-variable dataset for salt-affected cropland identification using VV + VH dual polarization, reflectance bands, and vegetation indices;

4.    Determine the best solution for mapping salt-affected cropland in dry and wet seasons using the overall accuracies and indicators from the confusion matrix of various datasets.

## 2. Materials and Methods

### 2.1. The Study Area

HID is located at the top of the northernmost Bay of the Yellow River and spans a region situated at 106°11′35″E–109°53′52″E and 40°10′30″–41°16′43″N. HID comprises five counties in Bayannur city with a total area of 17,243.23 km² (Figure 1), with 733,333.33 hectares of cropland. The crop yield has been stable at more than 5 billion tons for a long time. It is Asia's largest artesian irrigation area and one of China's three largest irrigation areas. In addition, HID was included in the World Irrigation Engineering Heritage List in 2019. Spring wheat, corn, vegetables, citrus, and sunflowers are the main crops in HID (see phenology of main crops in HID in Table S1). Vegetables are grown in a few places after the spring wheat harvest, while other crops are sown as one-season crops.

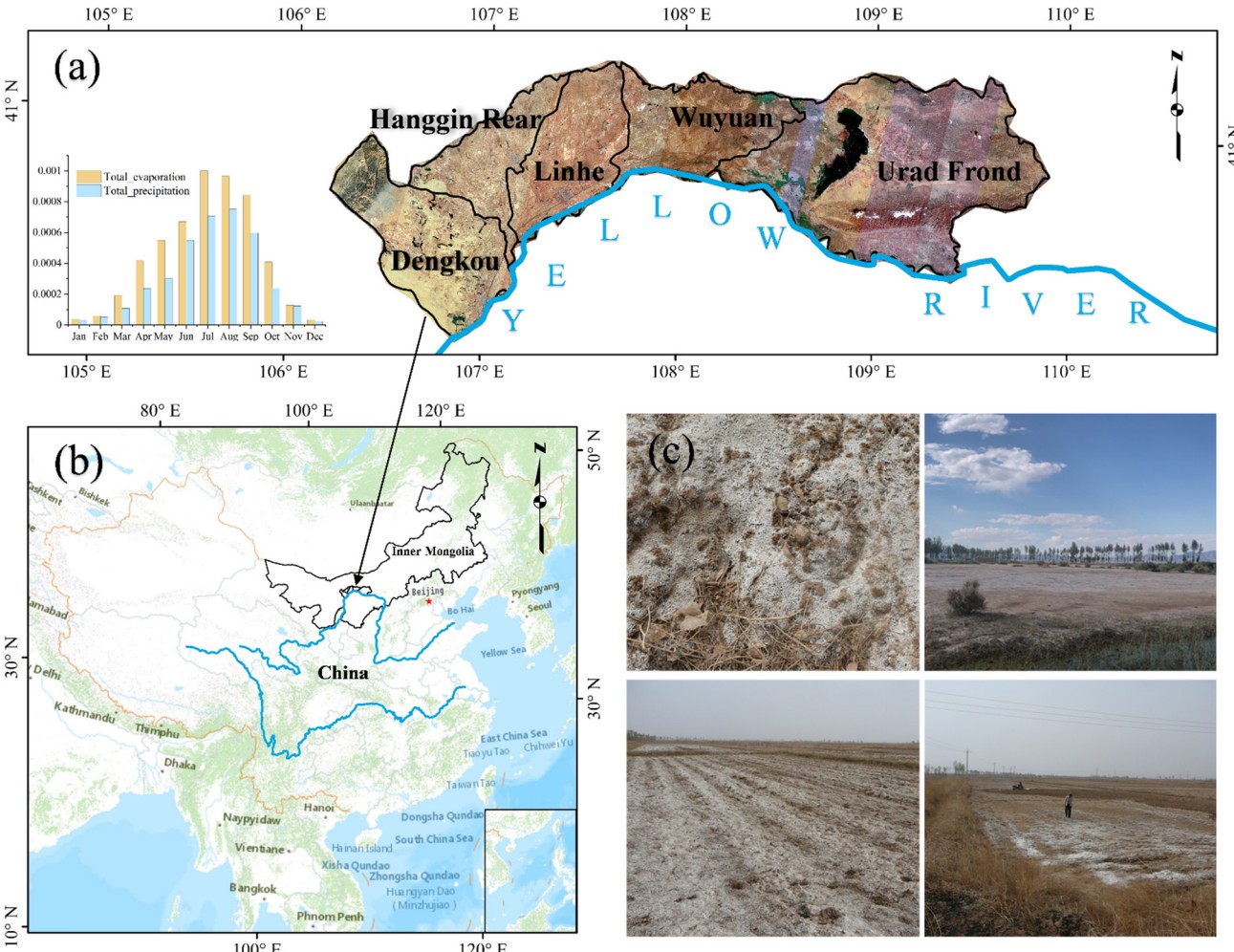

**Figure 1.** (**a**) Location of HID with histogram diagram of total evaporation and total precipitation in the last decade (meteorological data were obtained from ERA5_LAND data collection) and PlanetScope images acquired from 1 to 13 April in 2021 (shown in false color composited-R: NIR (Near-Infrared) band, G: Green band, B: Blue band). (**b**) The location of HID in China and the Inner Mongolia Autonomous Region. (**c**) Field photographs taken on April 2021, showing the salt on the soil surface before crops were planted.

A typical temperate continental climate prevails in the study area. The number of hours of sunshine per year is 3210.8–3305.8; the total amount of solar radiation is 146–152 kcal per square centimeter; the average yearly temperature is 6.1–7.6 °C; the daily average temperature difference is 13–14 °C. Additionally, the average annual evaporation is 2200 mm, which is nearly twelve times the average annual precipitation of 180 mm.

*2.2. Data*

2.2.1. Filed Sampling

The sample data for mapping saline cropland are based initially on the ground survey samples in previous studies. In contrast, the PlanetScope images acquired from April 2021 were utilized for delineating the reference samples using a visual interpretation strategy in this study, which was mainly because of the controlling measurements for preventing the COVID-19 epidemic during the critical period for collecting the ground truth samples. To accurately distinguish salt-affected soil from non-salt-affected soil on the cropland base map of HID, 1000 saline samples and 1000 non-saline samples were selected, as shown in Figure 2. High-resolution PlanetScope images were used as a reference to assess whether the soil was saline or non-saline, and each sample was labeled as either salinized or non-salinized, following the principle introduced in Figure 1. The study area is a typical arid irrigation farming area in northern China, with no winter crops grown throughout the year. This means the cropland surface in the irrigation area is bare outside from the previous year's harvest to the sowing of the following year. Therefore, the soil salinity is in layers 0–10 cm from March to April, which means the salinity of soil is on the surface and can be distinguished by the naked eye. This phenomenon leads to the surface reflectance of the saline soil captured by the imagery is also significantly different from that of the healthy soil.

Moreover, to ensure the accuracy of the artificially delineated saline and non-saline cropland samples, the number and spatial distribution must be as consistent as possible. Therefore, after the initial sample data are selected, the validity of the samples needs to be checked to ensure that the samples can adequately represent the category to which they belong. In this part, the quantile and quantile plots testing method will be applied to validate whether the selected saline and non-saline samples obey the normal distribution (Section 2.4).

2.2.2. Remote Sensing Data Collection

- PlanetScope

PlanetScope, operated by Planet, is a constellation of approximately 130 satellites that is able to image the entire land surface of the earth every day (a daily collection capacity of 200 million $km^2$/day). PlanetScope images have a resolution of about 3 m per pixel. The four-band frame imager with a butcher-block filter provides Blue, Green, Red and NIR bands. The PlanetScope Ortho Scene Level 3B Product has been used for selecting samples visually since it is an orthorectified, scaled Top of Atmosphere (TOA) Radiance Surface Reflectance image product suitable for analytic and visual applications.

- Sentinel

The European Space Agency (ESA) was renamed the EU Global Security Monitoring GMES as the Copernicus program, considering service duplication and discontinuity in 2012. Sentinel satellites are part of the Copernicus program. Sentinel-1 and Sentinel-2 are two Earth observation satellites currently in service with high-resolution sensors that can be shared globally.

The Sentinel-1 mission consists of a constellation of two polar-orbiting satellites, Sentinel-1A and Sentinel-1B, operating day and night to perform C-band synthetic aperture radar imaging. SAR data with a 10 m resolution are available for 12 days revisit period. Commonly used Class 1 products include Single-Look Complex (SLC) and Ground Range Detection (GRD) products. SLC products preserve phase information and process at natural

pixel spacing; GRD products incorporate detected amplitudes and multi-look to reduce speckle effects. Currently, only GRD products with Sentinel-1 data are integrated with GEE. The Sentinel-1 SAR imagery, in the Interferometric Wide (IW) mode, C-band, with dual polarization VV and VH, was acquired from 1 March 2020, to 31 April 2020, in coincidence with the field samples' selection period.

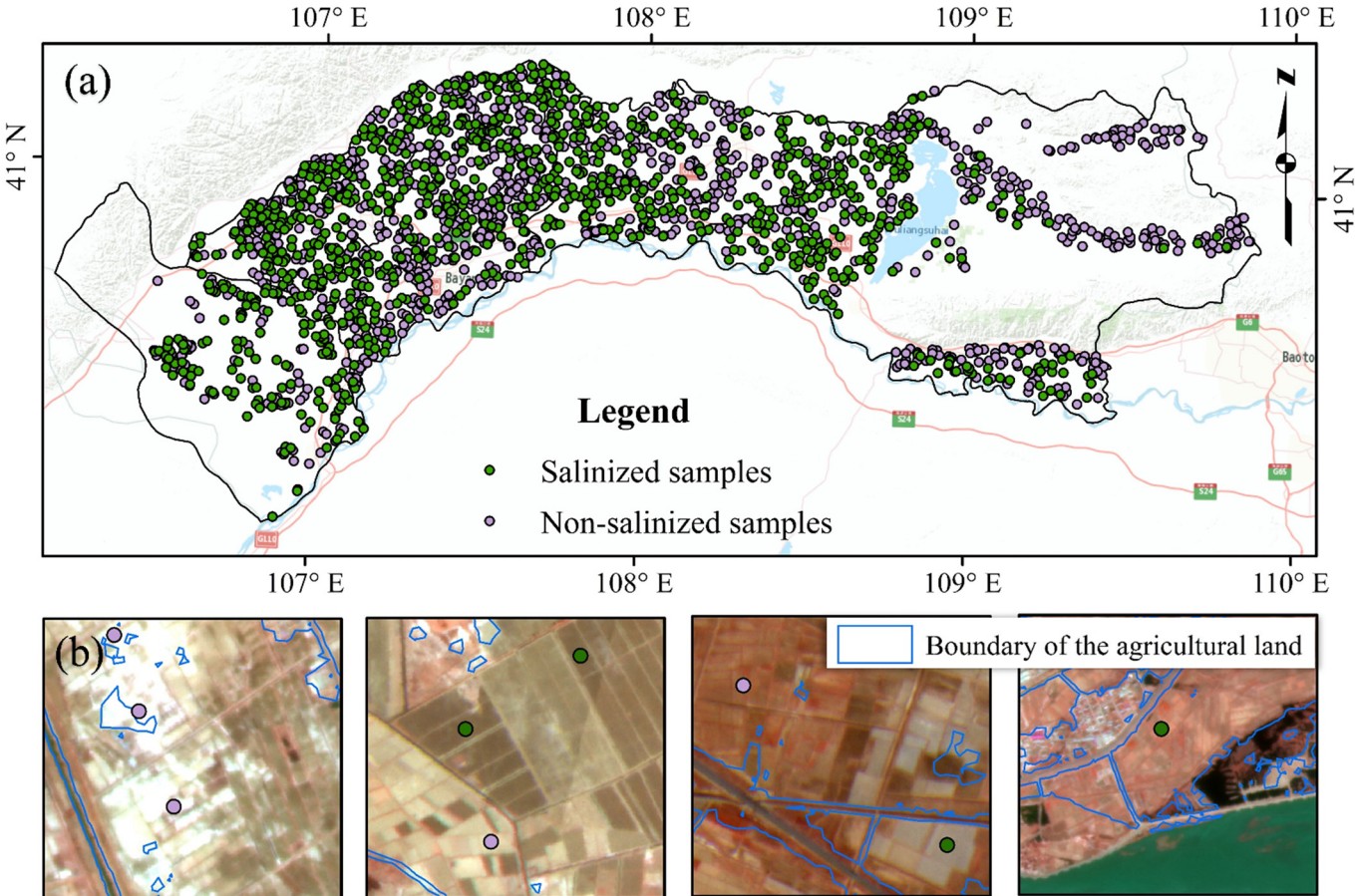

**Figure 2.** (**a**) The distribution of the samples of saline and non-saline cropland, (**b**) the samples in the false color composited PlanetScope image (R: NIR band, G: Green band, B: Blue band).

The Sentinel-2 mission consists of two solar polar-orbiting satellites, Sentinel-2A (23 June 2015–present) and Sentinel-2B (7 March 2017–present), distributed in a sun-synchronous orbit, each other into a 180° phase. Currently, Sentinel-2 mainly provides two product data: L1C and L2A. The L1C product is the reflectance data of TOA after orthorectification and sub-pixel geometric precision correction; the L2A product is the surface reflectance data product obtained using the Sen2cor tool officially provided by ESA to perform atmospheric correction on L1C. Data are available across Europe from October and globally from January 2017. Each Sentinel-2 satellite carries a multi-spectrometer MSI with 13 bands in the Visible, NIR, Narrow NIR and Short-Wave Infrared (SWIR) spectral ranges, including three Red Edge bands. Sentinel-2 Leve-1C and Leve-1A data products have been integrated into GEE (find details in Table A1). Considering that the Leve-1C product has more extended data availability, the Leve-1C TOA data product of Sentinel-2 has been selected for mapping saline cropland.

### 2.2.3. Ancillary Data

An initial cropland base map was created using the ESA WorldCover global land cover data package, which was developed based on the Sentinel-1 and Sentinel-2 at 10 m

resolution, and it may be accessed at https://viewer.esa-worldcover.org/worldcover, accessed on 16 October 2022. The cropland category is 40 in the ESA WorldCover global land cover data.

### 2.3. Generating the Cropland Base Map

In this section, the roads and irrigation ditches with a resolution of 2.4 m provided by AutoNavi Electronic Maps will be used to mask out the non-cropland areas within the fields of HID. Firstly, the GDAL module of Python extracted the roads and irrigation ditches from the electronic maps and then converted them to the SHP file. Secondly, the RASTERIO in Python was captured to mask the non-cropland parts from the WorldCover global cropland cover data.

### 2.4. Quantile and Quantile Plots Testing

Generally, the same type of ground objects should have the same or similar spectral reflectance characteristics in the same wavelength range of remote sensing images. The saline soils in arid regions mainly contain salts such as chlorides, sulfates and carbonates. Before the first irrigation of spring sowing in the Yellow River irrigation district, the salinity in the topsoil of 0–10 cm would be at the highest level, and saline elements would cover the soil surface, whitening the soil surface, as shown in Figure 1c. Spring wheat is the earliest sowing crop in the study area that cannot be grown in saline soil. Other crops, such as vegetables, corn and fruit, can be grown in soils with slight to moderate salinity. Sunflower is the main salt-tolerant crop and can even be planted in severe saline soil. Therefore, the 3 m resolution PlanetScope images obtained in early April (spring wheat grows in the Emergence Stage and can cover the ground surface) were chosen to collect sample data additionally to solve the problem that ground truth sample data are difficult to distribute evenly in a large area (Figure 2b).

It can be assumed that the eigenvalues of the saline and non-saline samples in different wavelength ranges obey the normal distribution. Conversely, when a specific sample contains anomalies, its distribution will deviate from the normal distribution. Therefore, the quantile graphical method (Quantile and Quantile Plot, Q-Q plot) can be used for sample validity tests for elements inconsistent during sample selection caused by visual interpretation errors. The Q-Q plot is a graphical technique for determining if two datasets come from populations with a common distribution. A Q-Q plot is the quantiles of the first dataset against the quantiles of the second dataset. Thus, the point (x, y) on the graph represents the quantile of the second dataset (y-coordinate) and the same quantile of the corresponding first dataset (x-coordinate). Therefore, the Q-Q plot will approximately lie on the line y = x superior if the two distributions are the same or similar. In this study, the x-axis was set as the normal data quantiles of the sample's reflectance value. In contrast, the y-axis was set as the normal theoretical quantiles to test whether the two categories of samples obey the normal distribution. The reflectance of the ten bands (B2, B3, B4, B5, B6, B7, B8, B8A, B11, B12) of the Sentinel-2 images observed from March to April 2021 (reduced to mean value on Google Earth Engine) are set as examples to illustrate the Q-Q plot (find the testing results in Section 3.2).

### 2.5. Modeling Strategy

The technical frame of this study is illustrated in Figure 3. First, the reflectance bands of Sentinel-1 and Sentinel-2 images were selected via spatial resolution to ensure the generalization and robustness of the models. In this step, bands at 60 m resolution were dedicated primarily to detecting atmospheric features and therefore are not included in subsequent research. Thus, indices and texture variables based on spectral reflectance were created at 10 m resolution. On the other hand, the backscattering signal of the Sentinel-1 VV + VH dual-polarization also participated in the modelling process at 10 m resolution.

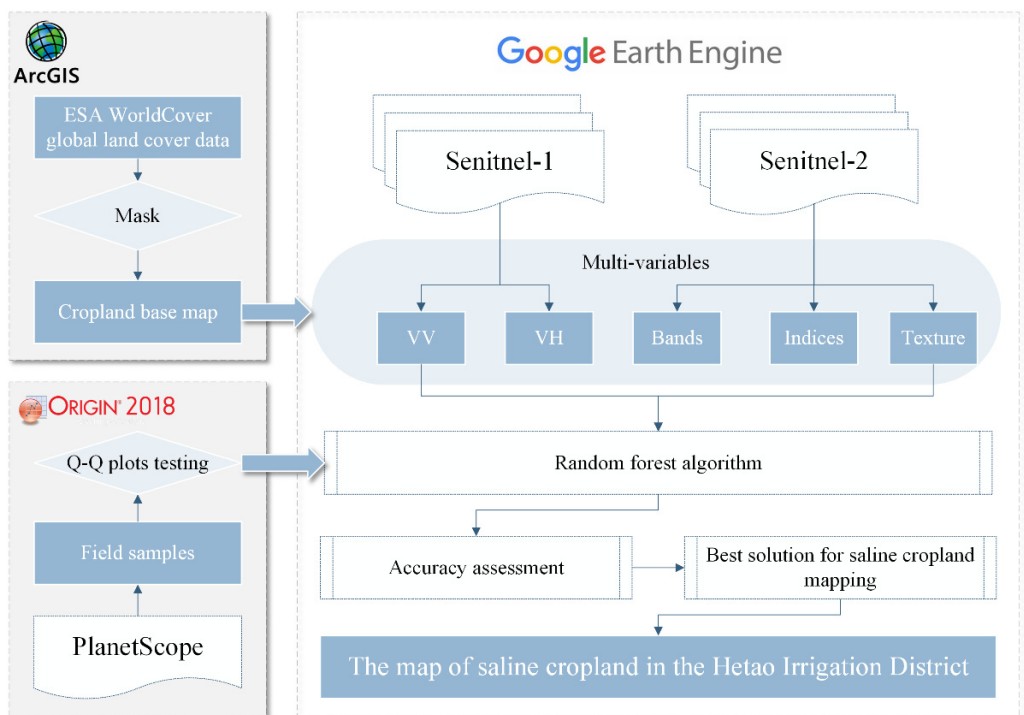

**Figure 3.** Technical framework of this study.

### 2.5.1. Spectral Salinity Indices

The wide range of wavelengths of the Sentinel-2 data has an excellent capability for remote sensing monitoring and mapping requirements of soil salinity [25]. Therefore, the mean values of Sentinel-2 spectral reflectance were included in the combined dataset to map the saline cropland in HID accurately. Table A1 (Section 2.2.2) lists the bands used in this study.

Applying spectral indices to investigate cropland salinity is built upon the different spectral behavior associated with image pixels of the ground object [43]. The salinization can dramatically change soil surface characteristics, leading to a significant difference from healthy soil, especially during the best monitoring period before the growing season in the arid and semi-arid regions with low vegetation cover and more exposed soil.

Moreover, the presence of salinity-tolerant crop coverage on the soil may also be a marker to reflect the soil salinization, thus allowing indirect mapping of salinity-stressed cropland [44]. On the other hand, unhealthy vegetation photosynthetic activity resulted in increased visible reflectance and decreased near-infrared reflectance (NIR) [25]. Therefore, several vegetation indices (VIs), such as Normalized Vegetation Index (NDVI), Soil-Adjusted Vegetation Index (SAVI), Optimized Soil Adjusted Vegetation Index (OSAVI) and Modified Soil-Adjusted Vegetation Index (MSAVI), were used to map soil salinity.

Numerous academics have regarded the VIs performance as appropriate for estimating soil salinity using remote sensing images [45]. To create a multi-variable model to map the saline cropland in an arid and semi-arid area, a succession of VIs commonly used for monitoring soil salinity was proposed in this study. Corresponding to this, other researchers have created various salinity indices, including the Normalized Difference Salinity Index (NDSI) and Salinity Index (SI), to identify and map soil salinity. Table A2 provides specific information. On the other hand, for combinations of two or three wavelengths in remote sensing images, extensive information can be obtained from the indices determined by spectral reflectance at the 10 m resolution.

### 2.5.2. Gray Level Co-Occurrence Matrix

Texture variables can provide valuable spatial information, reflecting the spatial distribution of the gray levels of remote sensing images and representing the spatial relationship between image features and the surrounding environment [46]. For instance, soil salinization in HID refers to the phenomenon in which the salt in the bottom soil or groundwater rises to the surface with capillary water. After the water evaporates, the salt accumulates in the surface layer. Thus, this phenomenon could significantly change the texture features of the land surface.

The textures are essential for identifying objects or regions of interest, whether in photographs, aerial photos, or satellite images. GEE provides the Gray Level Co-occurrence Matrix (GLCM) function to calculate broad applicability textures and can be utilized in various image classification applications [47–49]. In this study, the 14 GLCM indicators proposed by Robert et al. [50] and four other indicators proposed by Conners et al. [51] were used to construct texture variables. The reflectance-based texture variables based on the B2 with the highest accuracy of the Sentinel-2 images were obtained in GEE for modeling the mapping strategy for saline cropland in HID.

### 2.5.3. Classifier and Accuracy Assessment

Random Forest is one of the machine learning algorithms widely used in land cover classification [52] and has been applied to the remote sensing monitoring research of saline cropland gradually [53]. Furthermore, the importance evaluation function of the variables of Random Forest can screen out the variable that contributes the most to classification. Therefore, it can support further research on soil salinity monitoring.

Random Forest can build a multi-layer decision tree and randomly select subsets and variables of training samples. The classification accuracy of the Random Forest classifier on the GEE platform uses the incremental step of 100 trees to reach the highest accuracy with 600 trees. In addition, 70% of random samples are used to train the classifier, and 30% of random samples are used to validate the accuracy of the saline cropland classification.

Overall Accuracy (OA), Producer Accuracy (PA), and User Accuracy (UA) were used to evaluate the performance of Random Forest classifiers on the GEE. OA is the ratio of the total number of correctly classified pixels to the total number of pixels (the total number of pixels in the ground reference samples). UA corresponds to the probability that a randomly selected pixel from the map is classified as correct in the ground reference samples. PA corresponds to the likelihood that the reference sample is correctly classified on the map. The Kappa coefficient was previously considered an indicator that can be used for consistency checks and to measure classification effects. However, Foody [54] points out that the Kappa coefficient is not a measure of accuracy but an agreement beyond chance. Hence, it is unnecessary and should not be used in typical remote sensing applications. Therefore, Foody [54] argues that researchers should abandon the Kappa coefficient as a measure of accuracy instead of per-class accuracy estimation and confusion matrices to evaluate machine learning classification accuracy. Based on this, the Kappa coefficient is not used as a criterion for assessing the accuracies in this study.

In addition, to test the robustness of the RF on GEE, the F1 scores (F1 = 2 × UA × PA/ (UA + PA)) of saline cropland and non-saline cropland were also calculated. The F1-score is the harmonic mean of producer and user accuracies. In studies where the classification samples are not perfectly balanced, the F1 score is a strong indicator for testing the stability of classification. The F1 score ranges from 0 to 1, with higher scores indicating better classification performance.

## 3. Results

### *3.1. Cropland Base Map*

The cropland base map without roads and irrigation ditches was generated through the two steps introduced in Section 2.3. As a result, the boundaries of fields are more prominent, and the problem of adhesion between field pixels in the study area is eliminated, as shown

in Figure 4. There are 887,938.39 hectares of cropland in the study area. In addition, the area calculated from remote sensing results was compared with the data of The Third National Land Survey of China in 2020; a minimal difference between the modified cropland area and that of in land survey was found, and the specific data are relevant in Table 1. Therefore, the cropland base map is reliable and can be a basis for consecutive research.

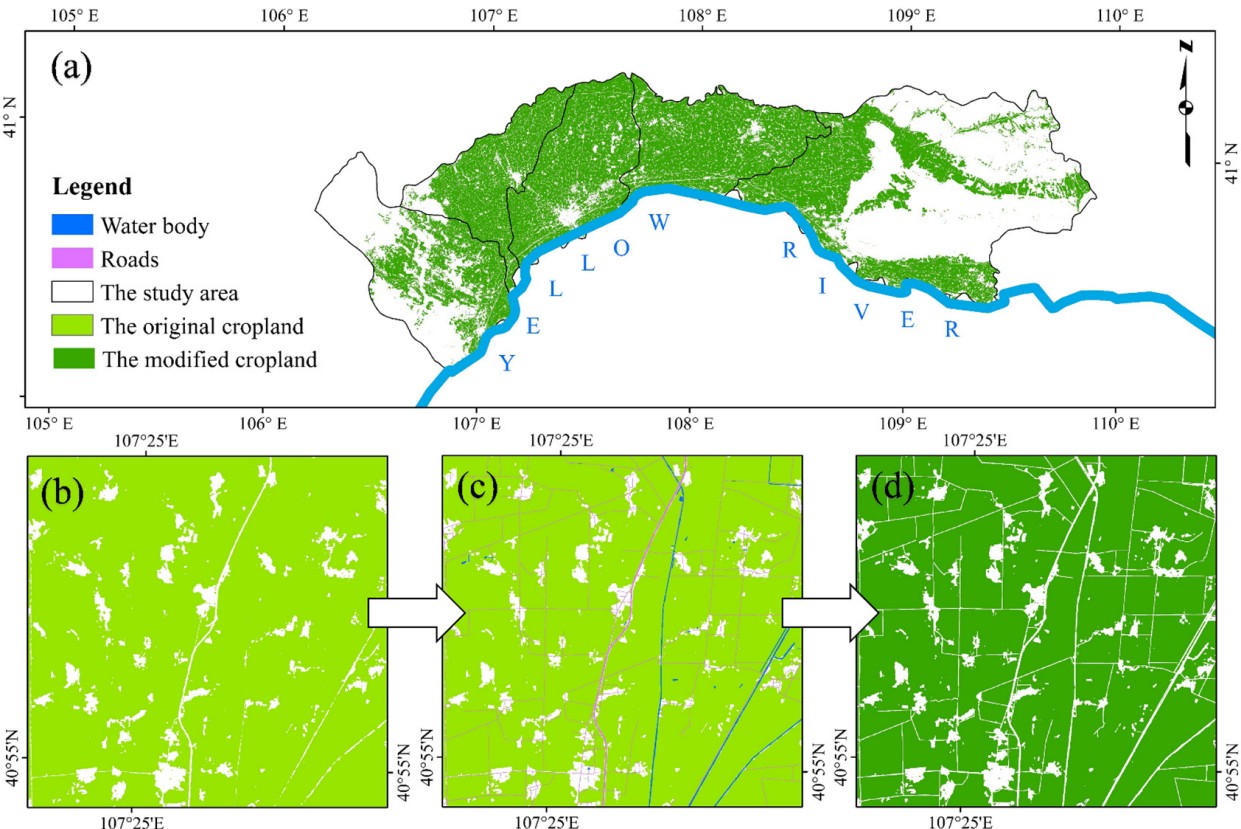

**Figure 4.** Cropland distribution in HID, (**a**) HID base map of modified cropland, (**b**) detailed map of initial cropland of ESA WorldCover global land cover data at 10 m resolution, (**c**) detailed SHP file of roads and ditches downloaded from AutoNavi Electronic Maps, (**d**) detailed map of modified cropland at 10 m resolution.

**Table 1.** Cropland area derived from remote sensing data and areas included in The Third National Land Survey.

| Counties | Modified Cropland Area (ha) | Cropland Area of the Third National Land Survey (ha) |
| --- | --- | --- |
| Dengkou | 110,074.32 | 96,931.29 |
| Hanggin Rear Banner | 135,266.37 | 111,651.51 |
| Linhe | 184,904.18 | 157,529.33 |
| Wuyuan | 199,808.2 | 171,927.30 |
| Urad Frond Banner | 257,885.32 | 231,604.35 |
| Total | 887,938.39 | 769,643.78 |

### 3.2. Sample Validity Test

The samples' pixel reflectance means values derived from the ten spectral bands of Sentinel-2 were acquired during the mapping period in this study (from March to April). Then, Origin 2018 was used to generate the Q-Q plot diagrams of the values of non-saline and saline cropland samples, as shown in Figure A1. In addition, the R-square ($R^2$) between the normal data quantiles and normal theoretical quantiles is a practical approach to showing the validity of samples. The specific $R^2$ values can be found in Table 2.

**Table 2.** $R^2$ values between normal data quantiles and normal theoretical quantiles of samples for the various bands of the Sentinel-2 image.

| Class | Non-Saline Cropland Samples | | | | | | | | | | Saline Cropland Samples | | | | | | | | | |
|---|---|---|---|---|---|---|---|---|---|---|---|---|---|---|---|---|---|---|---|---|
| Band | B2 | B3 | B4 | B5 | B6 | B7 | B8 | B8A | B11 | B12 | B2 | B3 | B4 | B5 | B6 | B7 | B8 | B8A | B11 | B12 |
| $R^2$ | 0.93 | 0.98 | 0.99 | 0.99 | 0.99 | 0.98 | 0.99 | 0.98 | 0.98 | 0.98 | 0.90 | 0.95 | 0.98 | 0.98 | 0.98 | 0.98 | 0.99 | 0.99 | 0.96 | 0.97 |

It can be found that the samples of either non-saline or saline cropland obey the normal distribution in the validation results. The scatter points in the Q-Q diagram of the saline and non-saline cropland before the growing season (from March to April) tend to fall on the x = y reference line, and the $R^2$ of all sample data is above 0.9. Whereas, the $R^2$ values of both non-saline and saline cropland samples on the B2 appear lowest (0.93 and 0.90, respectively) in all bands. This is because salt-affected soil has a valley of absorption close to the blue wavelength. Because of this, the reflectance is also lower than at other wavelengths. The sample points on the other bands in Figure A1 lie on the line x = y except for the B2 band, demonstrating the linear relationship between the normal data quantiles and the normal theoretical quantiles. This indicates a high level of sample consistency between two distributions of sample data on the B3 to B12 (Green to SWIR2) bands.

The results of the sample validity test show that the samples chosen in this study for two cropland classes before the growing season have adequate consistency and representativeness to meet the needs of the subsequent research.

### 3.3. Accuracy Assessment of Saline Cropland Mapping

The other probability of mapping saline cropland in the dry or wet season was tested in this part. According to observing the total precipitation in the last decade (chart in Figure 1a), the precipitation peaks in August and September and has been set in the wet season, while there was less precipitation from March to April and can be set as the dry season. Therefore, the accuracies of each dataset, including (1) the Sentinel-1 dual-polarization VV + VH dataset, (2) textures of Sentinel-2 B2 band, (3) Sentinel-2 spectral band dataset, (4) indices built based on Sentinel-2, and (5) the dataset of combined Sentinel-1 and Sentinel-2 in different time intervals were assessed to present the performance of the modeling strategies in the dry and wet season. The box plot of validation results are presented in Figure A2, and the validation indicators are shown in Table A3.

The results showed that the highest accuracy was achieved In the dry season from March to April, which was significantly higher than other time interval combinations. In comparison, no significant difference has been observed in the box plot of March, August and August to September. The result indicates that the two-month data combination in the dry season is the best solution for mapping saline cropland in HID. Thus, the dataset combined with March and April generated the salt-affected cropland map. Moreover, to clarify the best multi-variables with the highest accuracy for saline cropland mapping, the performances of the variables and their combinations were estimated, respectively, as shown in Figure 5 and Table A4.

As shown in Figure 5 and Table A4, the B2 band reflectance showed the highest accuracy (OA = 0.80) in the spectral bands of the Sentinel-2 data, which was followed by the B3 band. The Red Edge wavelength range is considered sensitive to green plants' growth status [55]. Thus, the B5, B6 and B7 bands of Sentinel-2 data showed no obvious advantage for salt-affected soil mapping at 20 m resolution in this case, as shown in Figure 5. Notably, the visible band has higher advantages for identifying salt-affected soil than the Infrared band in the dry season.

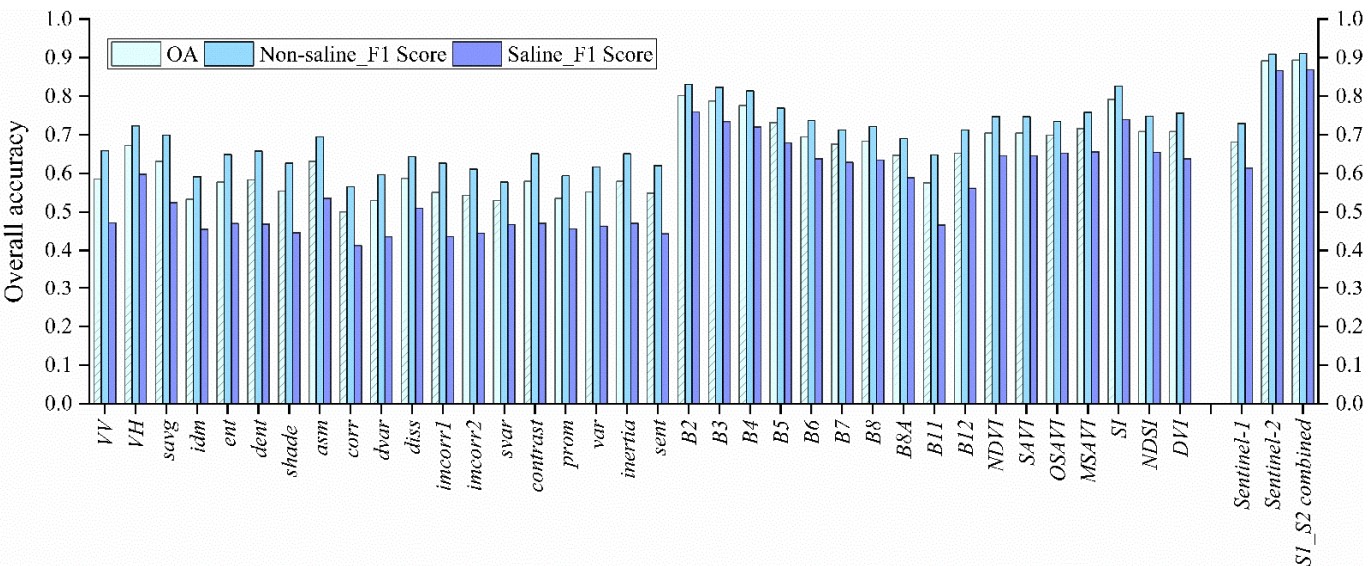

**Figure 5.** Classification accuracies of each variable and the combined datasets: Sentinel-1 indicates the mean value of Sentinel-1 dual-PolSAR (VV + VH) bands; Sentinel-2 suggests the combination of texture and indices derived from reflectance bands of Sentinel-2 and original mean values of spectral bands. S1_S2 combined indicates the combination of Sentinel-1 and Sentinel-2 datasets.

The VV + VH dual-polarization backscattering signal of Sentinel-1 data did not show competitive accuracy assessment results, with an overall accuracy of 0.68. The accuracy of the VH backscattering signal was higher than the VV backscattering signal, reaching 0.67 and 0.58, respectively. In addition, it can be seen in Table A4 that a very slight improvement (0.0019) has been detected in the assessment results by adding Sentinel-1 SAR data to Sentinel-2 spectral data. In this case, the mapping of salinity-affected crops is not significantly impacted by the VV + VH dual-polarization backscattering information. SAR data, however, can also be an essential supplemental data source in overcast and rainy regions where continuous optical images are challenging to obtain.

Each index variable's OA was greater than 70%, which denotes high accuracy. With an OA of 0.79, SI had the highest accuracy of any index, which was followed by MSAVI with an OA of 0.72. The degrees of accuracy for NDVI, SAVI, OSAVI, NDSI, and DVI are equivalent. These findings suggest that SI is the most appropriate indicator for saline cropland in salt-affected cropland mapping before the growing season in an arid and semi-arid region.

In this situation, combining Sentinel-1 and Sentinel-2 (S1_S2 combined in Figure 5), modeling strategies provided the optimal solution for saline cropland mapping, with an OA of 0.8938.

### 3.4. The Map of Saline Cropland

The dataset, including the spectrum reflectance, indices, texture information and PolSAR backscattering signal, produced the highest overall accuracy and F1 score (non-saline cropland is 0.91 and saline cropland is 0.87). Therefore, the Sentinel-1 and Sentinel-2 combined datasets were selected for mapping the saline cropland before the growing season in HID. Furthermore, the cropland in HID was classified into two categories, as aforementioned.

As seen in Figure 6, the amount of non-saline cropland in HID is more than the area of saline cropland, with 58.30% and 41.70% of cropland, respectively. The stretch between Wuyuan County to the west bank of Ulansuhai Nur is the largest saline cropland zone in HID. Additionally, the concentration of saline soil increases with proximity to the Yellow River Basin. As seen in Figure 6, the cropland is generally dispersed in strips toward the south and north to the east of Ulansuhai Nur. The majority of the cropland in the north uses drip irrigation, and some portions are watered by groundwater.

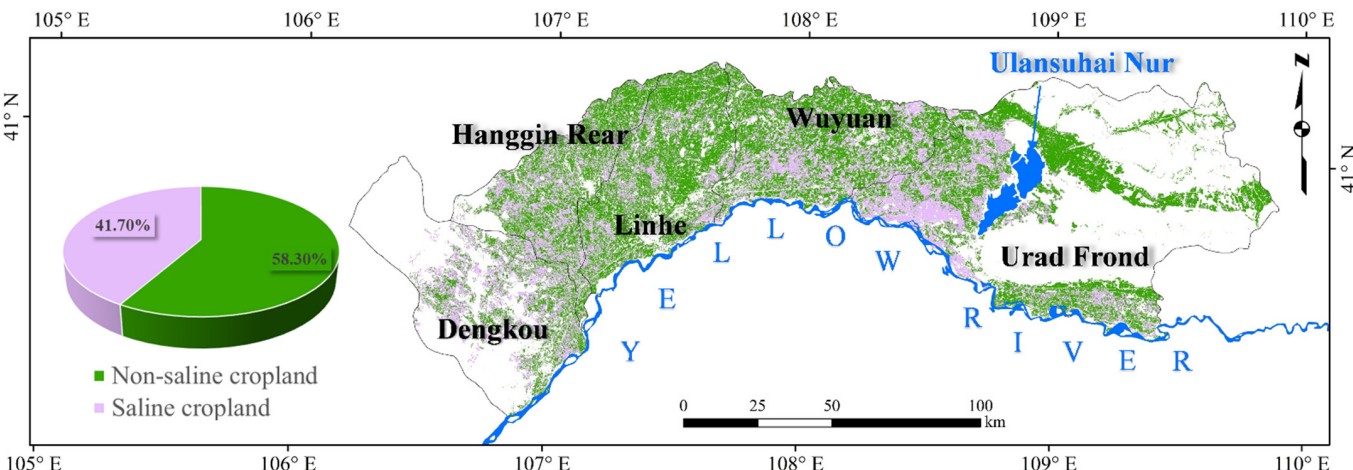

**Figure 6.** The distribution of non-saline and saline cropland with the percentage pie chart.

In contrast, the salinization is relatively high in the area irrigated mainly by the Yellow River in the south. As a result, there is less salinization than in the southern region. Numerous studies have shown that flood irrigating with Yellow River water causes soil salinization in HID. Therefore, the higher salinization is in keeping with the actual situation in areas irrigated with water from the Yellow River.

## 4. Discussion

### 4.1. Indices in Salt-Affected Cropland Mapping

Index variables were important in previous research on salt-affected soil monitoring and inversion. The analysis based on SI-MSAVI is the most renowned among them and has been shown to invert soil salinity [56–58] accurately. Likewise, NDVI and DVI, commonly used to monitor vegetation status, are also widely used in land salinization monitoring research and are critical indicators [59–61].

A mapping methodology for salinized cropland was developed in this study using several variables based on two bands (see Table A2 for details). The Red and NIR bands produce all other indices besides the SI. Figure 7 shows that even though the saline and non-saline samples have clear absorption valleys in the visible wavelength range, their reflectance values significantly differ. In comparison, the reflectance value in the NIR wavelength range is relatively high, but no clear difference has been observed. Near the two bands of water vapor (945 nm) and cirrus (1375 nm) of the Sentinel-2 image, there are more wide absorption valleys but practically overlapping curves in the SWIR wavelength range; however, near the SWIR1 and SWIR2 bands, the difference becomes more evident. Nevertheless, when employing a single SWIR band for accuracy assessment, the result does not achieve the high accuracy of the visible band due to the SWIR band's resolution of 20 m.

Commonly, SI measures the direct relationship between Electrical Conductivity (EC) and moisture. This ratio shows the salinity concentration in the available water [62]. By utilizing the more pronounced differences between the two cropland sample types in the Blue and Red bands, the SI index based on the visible band in this study was the variable with the highest contribution and achieved higher classification accuracy. However, other indices have similar classification accuracy since they are both constructed from red and NIR bands.

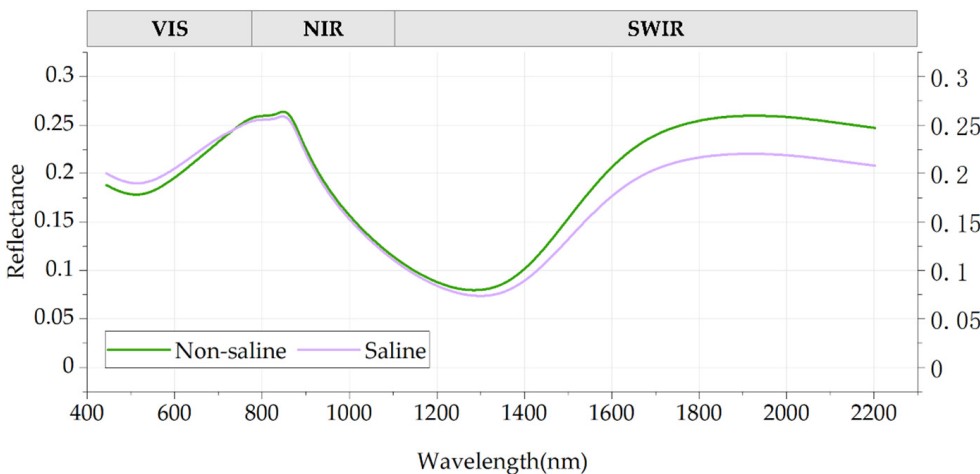

**Figure 7.** Reflectance curve of non-saline and saline cropland samples over the wavelength range of Sentinel-2 data in dry season.

While NDVI and NDSI represent normalized differences between the Red and NIR bands of the Sentinel-2 image, NDVI is the NIR minus the Red and NDSI is the inverse. Thus, a positive value of NDVI and a negative value of NDSI at the same pixel are equivalent. However, when NDVI or NDSI are not employed, the overall accuracy of the salt-affected cropland mapping slightly decreases (the accuracy decreases by 0.0019 when NDVI is removed, and the accuracy drops by 0.0058 when NDSI is removed). Consequently, NDVI and NDSI have equal correlation coefficients with the sample data, which means positive correlation coefficients for NDVI and negative correlation coefficients for NDSI). Additionally, NDSI was found to be more sensitive for detecting saline cropland in the wet season with OA at 0.66, which is slightly higher than the accuracy that NDVI can achieve in the wet season with OA at 0.65.

### 4.2. Multi-Sensor Data Application in Saline Cropland Mapping

Soil salinization is a severe problem faced by land worldwide, and the affected area is vast [18]. However, there is no exact standard for monitoring solutions due to different data sources and statistical methods. Unlike non-salt-affected land and other ground features, soil salinization has distinct and unique spectral reflectance characteristics and tends to show higher reflectance on spectral images [40,63–65]. Satellite remote sensing technology has irreplaceable advantages (near real-time and covering a large area) and good application prospects for observing soil salinity. Therefore, using multispectral remote sensing images to monitor the soil-affected cropland in an area with complex land surface objects is feasible.

On the other hand, microwave remote sensing has been widely used in the inversion of surface soil moisture and salinity for a long time [66,67]. Since the C-band polarization radar data of the Sentinel-1 satellite was introduced into civilian use, some breakthroughs have been made in soil moisture inversion research at the beginning [68,69]. However, the salinity change in the soil surface will affect the soil dielectric properties and thus will change the microwave emissivity of the land surface. Therefore, in addition to considering the impact of soil moisture alone, soil salinity has to be considered in areas with severe soil salinization [70–72]. As a result, the study of monitoring soil salinity using microwave remote sensing data has gradually attracted extensive attention [73,74].

The method combining the optical and microwave remote sensing data has been discussed preliminary in this study. However, many studies have shown that the identification ability of the backscattering coefficient will be significantly enhanced after the polarization decomposition of radar data. Nevertheless, the importance of radar data in this study is still minimal, which may be because the eigenvalues after polarization decomposition are more advantageous for identifying salt on the soil surface than the original backscatter-

ing coefficient. The GRD data provided in GEE do not have phase information, so it is impossible to realize GEE's polarization decomposition. Hence, it is difficult to establish the eigenvalues after polarization decomposition in a large area to extract saline cropland.

The application of remote sensing to earth observation is an essential means to understand the earth and study various natural phenomena in the future. Remote sensing technology is constantly developing, including many commercial satellite programs. As a result, the earth will be observed without a dead angle. In addition, the data volume will increase in geometric multiples; managing and using data efficiently and reasonably will be both a challenge and an opportunity for developing various algorithms and applications for salt-affected cropland monitoring.

*4.3. Strongly Saline Cropland Abandonment in HID*

The ESA WorldCover global land cover data did not recognize some fields with severe salinization as cropland. However, it is a minor error, because these have been abandoned for many years. On the other hand, a few severely salinized croplands have been planted late for sunflower seeds because of their salt tolerance [75]. In either case, it points to the severely salinized cropland in HID under the high potential abandonment stress.

Soil salinization has become an essential topic of global change research. The latest research shows that global soil salinization will be characterized by regional prominence, global intensification, and the coexistence of local salinization and intensification. Severely salinization is one of the most hazardous reasons why cropland is removed from production and then causes the abandonment globally of 0.3–1.5 million hectares per year [76]. It is generally recognized that a large proportion of salt-affected soils in irrigated areas occurs on land inhabited by smallholder farmers. However, salt-affected cropland degradation's social and economic dimensions have received little attention compared to its biophysical aspects [77].

Well-known examples of salt-induced land degradation include the Aral Sea Basin (Amu-Darya and Syr-Darya River Basins) in Central Asian countries, the Indo-Gangetic Basin in India, the Indus Basin in Pakistan, the Yellow River Basin in China, the Euphrates Basin in Syria and Iraq, the Murray-Darling Basin in Australia, and the San Joaquin Valley in the United States. Severe salinity also reduces paddy yields in many previously productive land areas; many paddy fields in Jaffna Peninsula, Sri Lanka, have been abandoned and are currently becoming shrubland [78]. Nevertheless, there has been no comprehensive study on the contribution of soil salinity to reduced agricultural productivity and the abandonment of paddy lands in a region. A study based on the analysis of the spatiotemporal variation in cropland expansion and loss in Xinjiang over 20 years found that the abandonment was the primary reason for the loss of cropland, with soil salinization playing an increasingly major role in the cropland abandonment [79]. Furthermore, Wu et al. [80] found widespread abandonment of reclaimed land and tillage in Xinjiang. A major reason for this abandonment was soil salinization with as much as 12,680 km$^2$ of cropland being affected.

There was a strong sense of expansion in the land use pattern of humanity with a poor understanding of sustainable development in the last few decades in Inner Mongolia. As a result, the saline bare land in northeast China has been utilized to a certain extent. However, due to the lack of protective technology, paddy fields' abandonment and salinization reappearance have also occurred in some areas after the high-intensity utilization of cropland. The extensive area saline cropland treatment was also implemented in 2022 with the government's support since the abandoned cropland is an essential reserve in China. The efficient utilization of salinized land is vital to ensure national food security, especially under the current COVID-19 pandemic and the global background of frequent disasters; it is imminent to utilize the reserve cropland and control the salinity.

## 5. Conclusions

In this study, after manual visual selection of samples, creation of a cropland base map of HID, and sample validity testing, a saline cropland identification model based on multi-sensor remote sensing data and the multi-variable dataset was built and achieved with high classification accuracy. One of these, the sample validity test method, was used for the first time in the saline cropland monitoring study and produced promising experimental results. The multi-variable dataset based on Sentinel-1 SAR and Sentinel-2 multispectral images from March to April furthermore exhibits strong performance in the remote sensing mapping of salt-affected crops. The methodology and results reported in this study may be advantageous for mapping saline cropland before the growing season in arid and semi-arid regions. They can therefore be encouraged and utilized in a broader area.

**Supplementary Materials:** The following supporting information can be downloaded at: https://www.mdpi.com/article/10.3390/rs14236010/s1, Table S1: Phenology of major crops in HID, Table S2: The accuracies of each dataset in the different time intervals during wet and dry season, Table S3: Classification accuracies of each variable and datasets for saline cropland mapping.

**Author Contributions:** Conceptualization, D.W. and A.H.; methodology, D.W., J.B. and L.G.T.C.; software, D.W.; validation, Z.S., R.C. and H.X.; formal analysis, D.W. and N.W.; investigation, J.B., J.P. and T.W.; resources, A.H.; data curation, D.W.; writing—original draft preparation, D.W.; writing—review and editing, L.S., D.W., L.G.T.C. and S.W.; visualization, Z.S., Q.X. and L.W.; supervision, L.G.T.C.; project administration, T.R.; funding acquisition, T.R. All authors have read and agreed to the published version of the manuscript.

**Funding:** This research was funded by Inner Mongolia Autonomous Region Science and Technology Plan Project (No. 2021GG0024) and The Introduction and Re-Innovation of The Japanese AgriLook System by Science and Technology Department of Inner Mongolia Autonomous Region.

**Data Availability Statement:** The earth engine code of the classification process of this study has been available on the website https://code.earthengine.google.com/148a08017e0f363c8b7414036a630313, accessed on 16 November 2022.

**Conflicts of Interest:** The authors declare no conflict of interest. The funders had no role in the design of the study; in the collection, analyses, or interpretation of data; in the writing of the manuscript; or in the decision to publish the results.

## Appendix A

**Table A1.** Spectral bands of Sentinel-2 MSI sensor for saline cropland mapping.

| Acronym | Bands | Spatial Resolution/m | Central Wavelength/nm |
|---|---|---|---|
| B2 | Blue | 10 | 496.6 (S2A)/492.1 (S2B) |
| B3 | Green | 10 | 560 (S2A)/559 (S2B) |
| B4 | Red | 10 | 664.5 (S2A)/665 (S2B) |
| B5 | Red Edge 1 | 20 | 703.9 (S2A)/703.8 (S2B) |
| B6 | Red Edge 2 | 20 | 740.2 (S2A)/739.1 (S2B) |
| B7 | Red Edge 3 | 20 | 782.5 (S2A)/779.7 (S2B) |
| B8 | NIR | 10 | 835.1 (S2A)/833 (S2B) |
| B8A | Narrow NIR | 20 | 864.8 (S2A)/864 (S2B) |
| B11 | SWIR 1 | 20 | 1613.7 (S2A)/1610.4 (S2B) |
| B12 | SWIR 2 | 20 | 2202.4 (S2A)/2185.7 (S2B) |

**Table A2.** Information list of reference spectral indices.

| Vegetation Index | Acronym | Formula |
|---|---|---|
| Normalized Difference Vegetation Index | NDVI | $\frac{R_{nir}-R_{red}}{R_{nir}+R_{red}}$ |
| Difference Vegetation Index | DVI | $R_{nir}-R_{red}$ |
| Soil-Adjusted Vegetation Index | SAVI | $\frac{(R_{nir}-R_{red})\times(1+L)}{R_{nir}+R_{red}}$ |
| Optimized Soil Adjusted Vegetation Index | OSAVI | $\frac{(1+0.16)\times(R_{nir}-R_{red})}{R_{nir}+R_{red}+0.16}$ |
| Modified Soil-Adjusted Vegetation Index | MSAVI | $\frac{2*R_{nir}+1-\sqrt{(2\times R_{nir}+1)^2-8\times(R_{nir}-R_{red})}}{2}$ |
| Normalized Difference Salinity Index | NDSI | $\frac{R_{red}-R_{NIR}}{R_{red}+R_{NIR}}$ |
| Salinity Index | SI | $\sqrt{R_{blue}\times R_{red}}$ |

R means the reflectance of spectral band of Sentinel-2 image, for example $R_{red}$ indicates the reflectance value of red band.

**Table A3.** The accuracies of each dataset in the different time intervals during wet and dry season.

| Datasets | Dry Season | | Wet Season | |
|---|---|---|---|---|
| | Mar | Mar to Apr | Aug | Aug to Sep |
| Sentinel-1 PolSAR | 0.6216 | 0.6815 | 0.5444 | 0.5946 |
| Textures of Sentinel-2 Blue band | 0.7722 | 0.7722 | 0.6583 | 0.6622 |
| Spectral bands of Sentinel-2 | 0.8012 | 0.8764 | 0.6680 | 0.7104 |
| Indices built based on Sentinel-2 | 0.8243 | 0.8687 | 0.6680 | 0.7104 |
| Combined dataset of Sentinel-1 and Sentinel-2 | 0.8610 | 0.8938 | 0.7220 | 0.7181 |

**Table A4.** Classification accuracies of each variable and datasets for saline cropland mapping.

| Bands | Overall Accuracy | User Accuracy | | Producers Accuracy | | F1 Score | |
|---|---|---|---|---|---|---|---|
| | | Non-Saline | Saline | Non-Saline | Saline | Non-Saline | Saline |
| VV | 0.5849 | 0.6571 | 0.4729 | 0.6592 | 0.4706 | 0.6582 | 0.4717 |
| VH | 0.6718 | 0.7400 | 0.5780 | 0.7070 | 0.6176 | 0.7231 | 0.5972 |
| savg | 0.6313 | 0.6916 | 0.5330 | 0.7070 | 0.5147 | 0.6992 | 0.5237 |
| idm | 0.5328 | 0.6295 | 0.4208 | 0.5573 | 0.4951 | 0.5912 | 0.4550 |
| ent | 0.5772 | 0.6537 | 0.4641 | 0.6433 | 0.4755 | 0.6485 | 0.4697 |
| dent | 0.5830 | 0.6551 | 0.4703 | 0.6592 | 0.4657 | 0.6571 | 0.4680 |
| shade | 0.5541 | 0.6361 | 0.4366 | 0.6178 | 0.4559 | 0.6268 | 0.4460 |
| asm | 0.6313 | 0.6977 | 0.5314 | 0.6911 | 0.5392 | 0.6944 | 0.5353 |
| corr | 0.5000 | 0.5979 | 0.3840 | 0.5350 | 0.4461 | 0.5647 | 0.4127 |
| dvar | 0.5290 | 0.6207 | 0.4123 | 0.5732 | 0.4608 | 0.5960 | 0.4352 |
| diss | 0.5869 | 0.6748 | 0.4784 | 0.6146 | 0.5441 | 0.6433 | 0.5092 |
| imcorr1 | 0.5502 | 0.6311 | 0.4306 | 0.6210 | 0.4412 | 0.6260 | 0.4358 |
| imcorr2 | 0.5425 | 0.6305 | 0.4260 | 0.5924 | 0.4657 | 0.6108 | 0.4450 |
| svar | 0.5290 | 0.6326 | 0.4213 | 0.5318 | 0.5245 | 0.5779 | 0.4672 |
| contrast | 0.5792 | 0.6548 | 0.4663 | 0.6465 | 0.4755 | 0.6506 | 0.4709 |
| prom | 0.5347 | 0.6308 | 0.4226 | 0.5605 | 0.4951 | 0.5936 | 0.4560 |
| var | 0.5521 | 0.6414 | 0.4386 | 0.5924 | 0.4902 | 0.6159 | 0.4630 |
| inertia | 0.5792 | 0.6548 | 0.4663 | 0.6465 | 0.4755 | 0.6506 | 0.4709 |
| sent | 0.5483 | 0.6325 | 0.4306 | 0.6083 | 0.4559 | 0.6201 | 0.4429 |
| B2 | 0.8012 | 0.8576 | 0.7265 | 0.8057 | 0.7941 | 0.8309 | 0.7588 |
| B3 | 0.7876 | 0.8312 | 0.7238 | 0.8153 | 0.7451 | 0.8232 | 0.7343 |
| B4 | 0.7761 | 0.8214 | 0.7095 | 0.8057 | 0.7304 | 0.8135 | 0.7198 |
| B5 | 0.7317 | 0.8028 | 0.6419 | 0.7389 | 0.7206 | 0.7695 | 0.6790 |
| B6 | 0.6950 | 0.7727 | 0.5991 | 0.7038 | 0.6814 | 0.7367 | 0.6376 |
| B7 | 0.6757 | 0.7704 | 0.5726 | 0.6624 | 0.6961 | 0.7123 | 0.6283 |
| B8 | 0.6834 | 0.7737 | 0.5820 | 0.6752 | 0.6961 | 0.7211 | 0.6339 |
| B8A | 0.6467 | 0.7365 | 0.5436 | 0.6497 | 0.6422 | 0.6904 | 0.5888 |
| B11 | 0.5753 | 0.6516 | 0.4615 | 0.6433 | 0.4706 | 0.6474 | 0.4660 |
| B12 | 0.6525 | 0.7147 | 0.5583 | 0.7102 | 0.5637 | 0.7125 | 0.5610 |
| NDVI | 0.7046 | 0.7766 | 0.6123 | 0.7197 | 0.6814 | 0.7471 | 0.6450 |
| SAVI | 0.7046 | 0.7766 | 0.6123 | 0.7197 | 0.6814 | 0.7471 | 0.6450 |
| OSAVI | 0.6988 | 0.7883 | 0.5984 | 0.6879 | 0.7157 | 0.7347 | 0.6518 |
| MSAVI | 0.7162 | 0.7831 | 0.6278 | 0.7357 | 0.6863 | 0.7586 | 0.6557 |
| SI | 0.7915 | 0.8344 | 0.7286 | 0.8185 | 0.7500 | 0.8264 | 0.7391 |
| NDSI | 0.7085 | 0.7860 | 0.6137 | 0.7134 | 0.7010 | 0.7479 | 0.6545 |
| DVI | 0.7085 | 0.7672 | 0.6244 | 0.7452 | 0.6520 | 0.7561 | 0.6379 |
| Sentinel-1 | 0.6815 | 0.7525 | 0.5874 | 0.7070 | 0.6422 | 0.7291 | 0.6136 |

**Table A4.** *Cont.*

| Bands | Overall Accuracy | User Accuracy | | Producers Accuracy | | F1 Score | |
|---|---|---|---|---|---|---|---|
| | | Non-Saline | Saline | Non-Saline | Saline | Non-Saline | Saline |
| Sentinel-2 | 0.8919 | 0.9272 | 0.8426 | 0.8917 | 0.8922 | 0.9091 | 0.8667 |
| S1_S2 combined | 0.8938 | 0.9274 | 0.8465 | 0.8949 | 0.8922 | 0.9109 | 0.8687 |

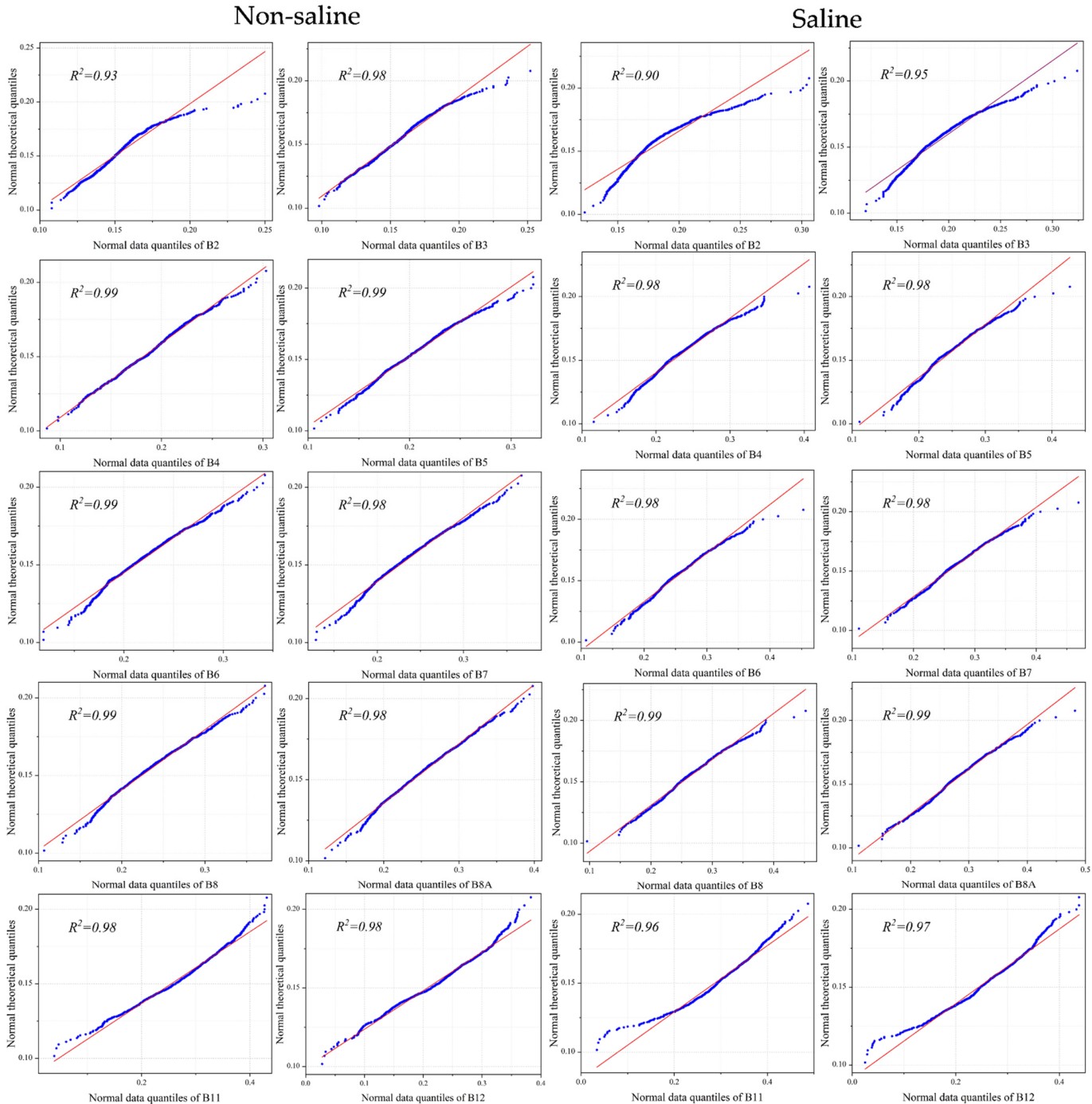

**Figure A1.** Q-Q test plots for samples' consistency of different classes on the mean values of the different bands of Sentinel-2 image, which reduced by mean value from March to April 2021.

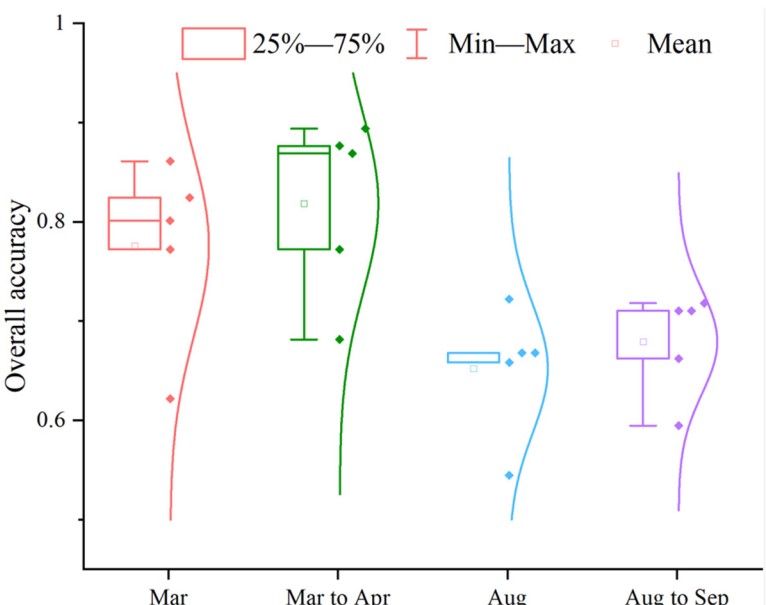

**Figure A2.** Results of the box plot accuracy assessment for several dataset combinations in the dry and wet seasons. The dry season in HID is represented by Mar and Mar to Apr on the x-axis, whereas the wet season in HID is represented by Aug and Aug to Sep on the x-axis.

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
