# Peer review of "Generating Salt-Affected Irrigated Cropland Map in an Arid and Semi-Arid Region Using Multi-Sensor Remote Sensing Data"

_remotesensing, doi:10.3390/rs14236010_

Round 1

Reviewer 1 Report

The manuscript is interesting and may be considered for publication after undergoing Major Revision. Below I send my thoughts.

- Abstract is poorly structured. It needs to contain the hypotheses and objectives clearly. Place the main methodological procedures.

- The introduction is well outlined, but tiring to read as it is too long. It needs to demonstrate what are the main gaps that exist and that this research fills. Put the main hypotheses and objectives clearly at the end of the Introduction.

- Why was only Random forest used? Why haven't other machine learning algorithms been tested? I believe this is something to be mentioned in the Introduction inclusive.

- There are some figures and tables that are not suitable for the manuscript and should be inserted as supplementary material (Table 2 and Figures 5 and 6).

- Discussion of results is very shallow and needs to be improved.

Author Response

Manuscript title: Generating Salt-affected Irrigated Cropland Map in an Arid and Semi-Arid Region Using Multi-Sensor Remote Sensing Data (ID: 1986981).

Dear editor,

We would like to thank you for the possibility of submitting the revised version of our manuscript. Also, we would like to thank the reviewers for their valuable comments and suggestions, which contributed to improving the manuscript's quality. All points raised by the reviewers have been addressed, and the responses are highlighted in blue and red in the manuscript.

We appreciate the contribution from the editor and reviewers and the assistance we are receiving during the reviewing process.

Replies to the reviewer’s comments:

Reviewer#1

  1. Abstract is poorly structured. It needs to contain the hypotheses and objectives clearly. Place the main methodological procedures.

Response: Considering the reviewer’s constructive comments, we restructured the introduction part, and the hypotheses and objectives were presented more appropriated in the revised manuscript. Line 26-50.

  1. The introduction is well outlined, but tiring to read as it is too long. It needs to demonstrate what are the main gaps that exist and that this research fills. Put the main hypotheses and objectives clearly at the end of the Introduction.
  2. Why was only Random Forest used? Why haven't other machine learning algorithms been tested? I believe this is something to be mentioned in the Introduction inclusive.

Response: Based on comments 2 and 3, we made some adjustments to the Introduction section of the initial manuscript. To remove verbose expressions and make the opinion clearer, the second and third paragraphs were combined into one paragraph. Line 80-95.

Furthermore, paragraph 6 emphasizes the significance and irreplaceability of Random Forest algorithms in monitoring and mapping salt-affected cropland. Line: 123-131.

In addition, the gaps in previous research that this study will fill are explained by summarizing the deficiencies in previous research. Line 137-144. The hypotheses and objectives are more clearly stated in the last two paragraphs of the Introduction section. Line 145-162.

  1. There are some figures and tables that are not suitable for the manuscript and should be inserted as supplementary material (Table 2 and Figures 5 and 6).

Response: Thank you for your constructive comments regarding the manuscript's structure. Table 2 and Figures 5 and 6 have been set into Appendix A and renamed their title into Table A1, Figure A1 and Figure A2. Line 652,657,662.

  1. Discussion of results is very shallow and needs to be improved.

Response: We have improved section 3.2 to deliver the mined information to readers more completely and clearly. Line 413-426.

The Discussion part has been revised fully considering the comments of reviewers 1 and 3 regarding the Discussion part. Therefore, we added a new section to the Discussion part to make the manuscript more fluent and logical by reorganizing the descriptions of several indices in Table 2 of the initial manuscript into the Discussion part. Line 504-544.

Thank you once more for the opportunity to submit the revised manuscript based on the constructive comments and helpful suggestions.

Kind regards,

Sincerely yours,

Deji Wuyun

E-mail address: 82101181154@caas.cn

Reviewer 2 Report

Accept in present form.

Author Response

Thank you so much for your support; it greatly benefits us.

Reviewer 3 Report

The paper deals with the use of remote sensing data for evaluating and predicting soil salinization in a wide-scale areas. Although based on the previously developed criteria used in remote-sensing and hyperspectral imaging, the paper provides new data on the evaluation of soil properties.

The structure of the paper can be improved by double-checking the information and exchanging the parts of the text between the sections and clarifying some points.

1. The Materials and Methods seems to be very heavy and containing the information that can be removed, sent to Appendices or, to the contrary, to Discussion.

- PlanetScope and Sentinel generic details can be shortened, organized as an Appendix or added to Supplementary

- Table 2 as an 'information list' seems to be out of place, especially with some discussion in the final column. In this form, it could be organized as an Appendix, the discussion could probably added to the Discussion section, and Table 2 should be shortened to the actually calculated indexes and their strict value for this study. 

- The word 'bands' is used ambiguously throughout the text, meaning both wavelength ranges, information channels and integral reflectances. It should be checked, and in Materials and Methods, all the variables, like Rs in Table 2 should be properly introduced to avoid confusion.

- Apart from Table 2, all other cases when Materials and Methods contains not a brief inline explanation but a full-fledged discussion should (almost the whole Section 2.5) be avoided, and this part of the discussion should be moved to Discussion section.

- The real differences in use of NDVI and NDSI values should be given, now, without comments it may result in confusion.

2. If the Sentinel band names are used by default, it should be named in the same manner in all the instances, both in the text and tables and figures/captions.

3. Please consider representing Figure 5 in a more information bearing way. In the present form it is of auxiliary value, can be represented as a short table and is not very thoroughly discussed. The present figure can be added as Supplementary.

4. Normalized Difference Salinity Index (NDSI) is named incorrectly in Table 2

5. There are many misspellings and grammar errors throughout the text, it should be double-checked.

Author Response

Manuscript title: Generating Salt-affected Irrigated Cropland Map in an Arid and Semi-Arid Region Using Multi-Sensor Remote Sensing Data (ID: 1986981).

Dear editor,

We would like to thank you for the possibility of submitting the revised version of our manuscript. Also, we would like to thank the reviewers for their valuable comments and suggestions, which contributed to improving the manuscript's quality. All points raised by the reviewers have been addressed, and the responses are highlighted in blue and red in the manuscript.

We appreciate the contribution from the editor and reviewers and the assistance we are receiving during the reviewing process.

Replies to the reviewer’s comments:

Reviewer # 3

  1. The Materials and Methods seems to be very heavy and containing the information that can be removed, sent to Appendices or, to the contrary, to Discussion.

  • PlanetScope and Sentinel generic details can be shortened, organized as an Appendix or added to Supplementary

Response: Table 1, which presents detailed information about the Sentinel-2 data, has been added to Appendix and renamed Table A1. Line 652.

The generic details of PlanetScope, which have only four bands and don't show as a table, are maintained in the main text.

  • Table 2 as an 'information list' seems to be out of place, especially with some discussion in the final column. In this form, it could be organized as an Appendix, the discussion could probably add to the Discussion section, and Table 2 should be shortened to the actually calculated indexes and their strict value for this study.

Response: Response: The description in Table 2’s last column in the initial manuscript has been reconstrued as a new Discussion section, and the Table 2 has been removed into Appendix A. Line 504-544.

  • The word 'bands' is used ambiguously throughout the text, meaning both wavelength ranges, information channels and integral reflectance. It should be checked, and in Materials and Methods, all the variables, like Rs in Table 2 should be properly introduced to avoid confusion.

Response: Thank you very much for noticing the problem and pointing it out. This specific comment is valuable for improving the manuscript. We have corrected the word 'band' to be misused according to its actual meaning. Therefore, distinction is made according to exact meaning in the article, except that the information channel of the remote sensing image is still written as 'band'. Line 269,282,299,303,310,312,313,459.

  • part from Table 2, all other cases when Materials and Methods contains not a brief inline explanation but a full-fledged discussion should (almost the whole Section 2.5) be avoided, and this part of the discussion should be moved to Discussion section.

Response: The final column of table 2 in Section 2.5 has been removed the Discussion part, and only some necessary background knowledge has been retained. Line numbers: 327-335. 

  • The real differences in use of NDVI and NDSI values should be given, now, without comments it may result in confusion.

Response: The explanation of the difference in the use of NDVI and NDSI has been presented with details in the first section of Discussion part. Moreover, this section's last paragraph elaborates on why these two variables were used in this study simultaneously. Line 534-544.

  1. If the Sentinel band names are used by default, it should be named in the same manner in all the instances, both in the text and tables and figures/captions.

Response: Thank you for your kind reminder. Since the specific band names of the Sentinel-2 image were explained in Line 249-251, we have used abbreviations for subsequent references to band names. This update can be found all over the revised manuscript. However, in the first section of Discussion part of the revised manuscript, to ensure that the discussion is consistent with the information in Figure 7, no acronyms are used in this section to convey the topics we discussed more clearly to the reader.

  1. Please consider representing Figure 5 in a more information bearing way. In the present form it is of auxiliary value, can be represented as a short table and is not very thoroughly discussed. The present figure can be added as Supplementary.

Response: The previous table was moved to Appendix A and renamed the title as Figure A1. Also, the figure has been replaced by a short table (Table 2) showing the R-squared values of Q-Q plot testing. Line 411-415, 657.

  1. Normalized Difference Salinity Index (NDSI) is named incorrectly in Table 2.

Response: This mistake has been corrected in both main text and the table. Line 331, 645.

  1. There are many misspellings and grammar errors throughout the text, it should be double-checked.

Response: Thank you for your feedback. The text has been double-checked by a native speaker, which has improved the manuscript's quality.

Thank you once more for the opportunity to submit the revised manuscript based on the constructive comments and helpful suggestions.

Kind regards,

Sincerely yours,

Deji Wuyun

E-mail address: 82101181154@caas.cn

Reviewer 4 Report

In this work the Generating Salt-affected Irrigated Cropland Map in an Arid and Semi-Arid Region Using Multi-Sensor Remote Sensing Data was proposed.

the work appears to be complete throughout, there is a good introduction, it is clear in its exposition of materials, methods and results, and finally it contains a good discussion section. 

Author Response

Thank you so much for your encouragement and comments; they mean a lot to us.

Round 2

Reviewer 1 Report

This manuscript can be accept